# Transient role of the middle ear as a lower jaw support across mammals

**Neal Anthwal[1], Jane C Fenelon[2], Stephen D Johnston[3], Marilyn B Renfree[2], Abigail S Tucker[1]\***

[1]Centre for Craniofacial and Regenerative Biology, King's College London, London, United Kingdom; [2]School of BioSciences, University of Melbourne, Victoria, Australia; [3]School of Agriculture and Food Sciences, University of Queensland, Gatton, Australia

**Abstract** Mammals articulate their jaws using a novel joint between the dentary and squamosal bones. In eutherian mammals, this joint forms in the embryo, supporting feeding and vocalisation from birth. In contrast, marsupials and monotremes exhibit extreme altriciality and are born before the bones of the novel mammalian jaw joint form. These mammals need to rely on other mechanisms to allow them to feed. Here, we show that this vital function is carried out by the earlier developing, cartilaginous incus of the middle ear, abutting the cranial base to form a cranio-mandibular articulation. The nature of this articulation varies between monotremes and marsupials, with juvenile monotremes retaining a double articulation, similar to that of the fossil mammaliaform *Morganucodon*, while marsupials use a versican-rich matrix to stabilise the jaw against the cranial base. These findings provide novel insight into the evolution of mammals and the changing relationship between the jaw and ear.

## Introduction

In non-mammalian vertebrates, the jaw joint is formed between the quadrate (or palatoquadrate) of the upper jaw and the articular part of Meckel's cartilage, a rod of cartilage that runs through the lower jaw (*Figure 1A*). This is known as the primary jaw joint. In mammals, this function is carried out by a new joint between the dentary and squamosal bones, known as the temporomandibular joint or TMJ in humans, and is referred to as the secondary jaw joint. In mammals, the bones of the original primary jaw joint have been incorporated into the ear and play a role in hearing (*Anthwal et al., 2013*). In addition to forming a joint with the articular as part of the primary jaw joint, the amniote quadrate also articulates with the cranial base. During the evolutionary transition that gave rise to mammals, the connection between the quadrate and the cranial base simplified (*Luo and Crompton, 1994*). The robust quadrate of reptiles moved from being attached to up to five separate skeletal elements, able to bear the mechanical force of feeding, to become the diminutive mammalian incus, suspended by a ligament from a single cranial base bone, the petrosal, in an air-filled cavity allowing sound transmission (*Kemp, 2005*; *Kielan-Jaworowska et al., 2004*). At the same time, Meckel's cartilage lost its permanent nature, separating the incus and neighbouring malleus from the rest of the jaw in adults (*Figure 1B*; *Anthwal et al., 2017*; *Urban et al., 2017*).

Early mammal-like reptiles had a permanent Meckel's cartilage and joints between the quadrate and articular (Q-A), and posteriorly between the quadrate and cranial base – similar to extant reptiles (*Figure 1A*). In mammaliaforms, such as *Morganucodon*, both a primary Q-A and a secondary dentary squamosal joint were present, in addition to a joint between the quadrate (incus) and the paraoccipital process of the petrosal (*Figure 1C*). This petrosal and incus joint precedes detachment of the middle ear from Meckel's cartilage in mammal evolution (*Luo and Crompton, 1994*). A connection between the future middle ear bones and the cranial base is therefore a feature of fossil

\*For correspondence:
abigail.tucker@kcl.ac.uk

**Competing interests:** The authors declare that no competing interests exist.

**eLife digest** The defining feature of all mammals is how the jaw works. Fish, reptiles and other animals with backbones have a lower jaw made of many bones fused together, one of which connects to the upper jaw. The lower jaw in mammals, however, is made of a single bone that connects with the upper jaw using a completely unique jaw joint. This new joint emerged as the ancestors of all mammals split from the reptiles around 200 million years ago. The bones that formed the original jaw joint ended up in the middle ear in mammals and switched to a role in hearing.

Nowadays, there are three types of mammals: the placentals, marsupials and monotremes (the egg laying mammals). In mice, humans and other placental mammals, the skeleton of the adult jaw joint forms in the embryo before birth. However, marsupials (such as kangaroos and opossums) and monotremes (platypuses and echidnas) are born at a much earlier embryonic stage, before the adult jaw joint has formed. It is therefore unclear how newborn marsupials and monotremes are able to move their jaws to feed on milk from their mother.

Anthwal et al. compared how the jaw develops in mice, opossums, platypuses and echidnas before and after the adult jaw joint becomes functional. The experiments showed that young echidnas, platypuses and opossums use their middle ear bones to articulate the lower jaw with the head before the adult jaw joint forms. In young opossums, the ear bones form a cushion to support the jaw. In juvenile platypuses a double joint is evident, with the ear bones forming a joint at the same time as the newly formed adult jaw joint, similar to the situation observed in fossils of mammal ancestors. The experiments also indicated that mice and other placental mammals may potentially use their ear bones to support the jaw before birth.

These findings shed light on why the ear and jaw have such a close connection in mammals. In humans, the ear and jaw bones are still connected by ligaments, explaining why trauma to the jaw joint can cause dislocation of the ear bones. Similarly, defects in the development of the jaw can impact the ear, such as in Treacher Collins Syndrome, where in some cases the jaw joint fails to form and the ear bones appear to try and take this role. Understanding how the ear and jaw evolved will help us understand why they look like they do and why a defect in one can have knock-on consequences for the other.

mammaliaforms. In extant mammals, the proposed homologue of the paraoccipital process is the crista parotica, which forms as a cartilaginous spur off the petrosal and is derived from neural crest cells, distinct to the rest of the petrosal and otic capsule, which are mesodermally derived (*O'Gorman, 2005*; *Thompson et al., 2012*). Modern mammals have separated the middle ear from the jaw in adults, and the ossicles (malleus, incus and stapes) are now suspended by ligaments from the cranial base to allow free vibration during sound transmission from the ear drum to the inner ear (*Figure 1B*). Paleontological evidence indicates that the evolution of the definitive mammalian middle ear (DMME) occurred at least twice, once in the lineage that gave rise to monotremes and once in the therian (marsupial and eutherian) mammals (*Meng et al., 2018*; *Rich et al., 2005*), while new developmental data suggests that the two groups of therian mammals may have each independently acquired the DMME (*Urban et al., 2017*). Here, we refer to eutherian mammals rather than placental mammals, as marsupials have a yolk-sac placenta (*Renfree, 2010*).

Marsupials (*Allin, 1975*; *Filan, 1991*) and monotremes (*Griffiths, 1978*), exhibit extreme altriciality, greater than is seen in any eutherian (*Werneburg et al., 2016*). This has profound consequences for early feeding as the bones that form the mammalian jaw joint, the dentary and squamosal, have not fully ossified by the time of birth/hatching. The dentary-squamosal joint forms prior to birth in eutherian mammals, and begins to function in the embryo (*Habib et al., 2007*; *Jahan et al., 2014*). In the mouse, gestation is approximately 20 days, with breakdown of Meckel's cartilage, to separate the lower jaw from the ear bones, following during early postnatal stages (*Anthwal et al., 2013*). In contrast, the opossum *Monodelphis* has a short gestation of just 13 days (*Keyte and Smith, 2008*), and is born before development of the dentary-squamosal articulation, which forms between 14 and 20 days after birth (*Filan, 1991*; *Maier, 1987*). Monotremes hatch out of the egg after 10 days post-oviposition (*Griffiths, 1978*). The formation of the dentary-squamosal joint in monotremes

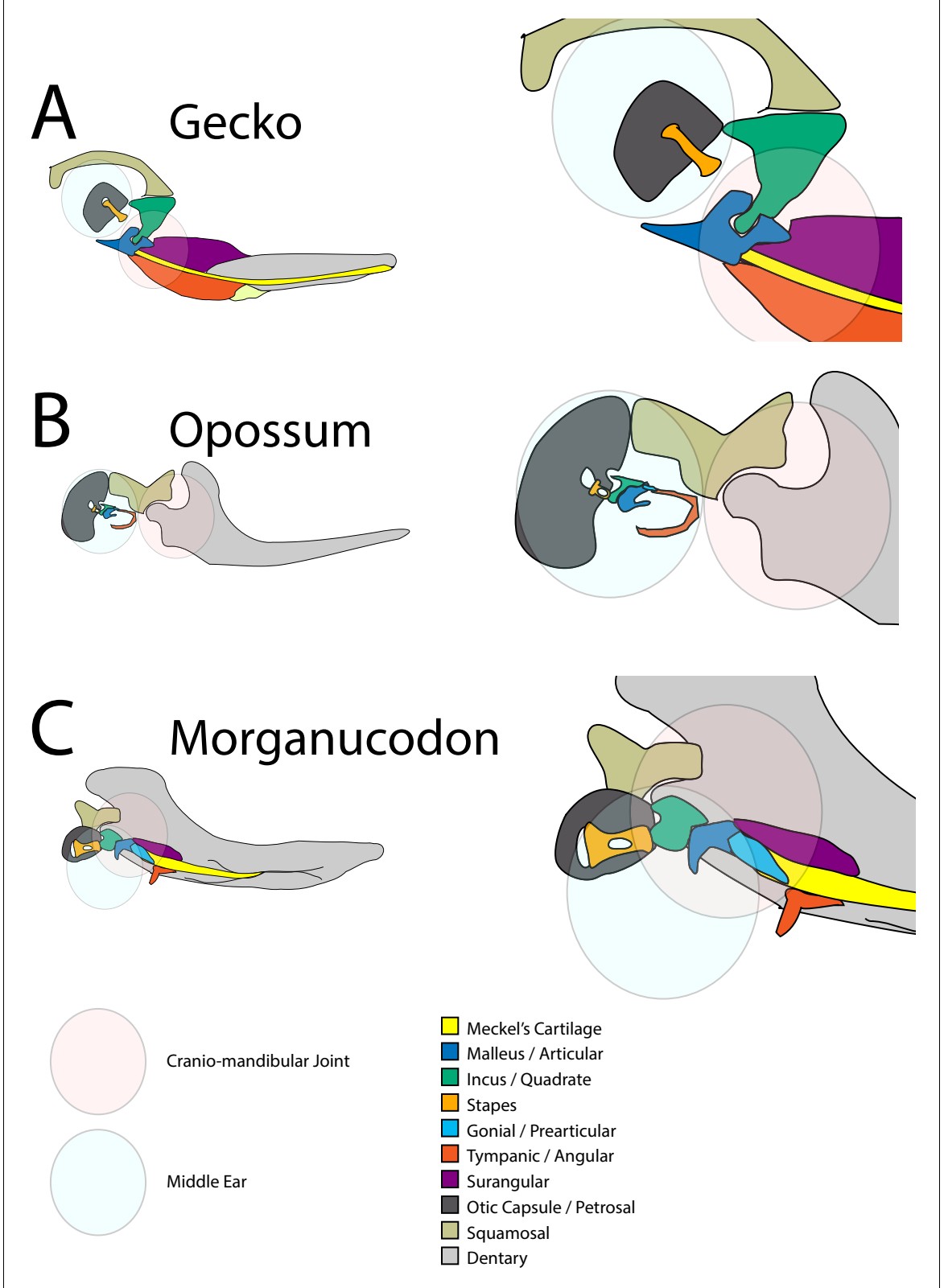

**Figure 1.** Schematic of cranial-mandibular jaw articulation showing the roles of the quadrate/incus and articular/malleus in the hearing and jaw joint modules in (A) reptile gecko, (B) mammal opossum, (C) mammal-like reptile Morganucodon.

has recently been followed and shown to form from 10 days after hatching in the platypus (*Anthwal and Tucker, 2020*). Breakdown of Meckel's cartilage in both marsupials and monotremes occurs relatively late postnatally (*Urban et al., 2017*), with a robust Meckel's still evident in nest young platypuses (*Zeller, 1993*). There is, therefore, a significant gap between birth and the advent of a functional mammalian jaw joint in both marsupials and monotremes.

The feeding strategies of new-born mammals vary in extant members of each group of mammals. Compared to eutherian mammals, marsupials rely on placental support for a relatively short period of time (*Renfree, 2010*) and consequently receive the nutrition required for their development via a lengthy and sophisticated lactation (*Tyndale-Biscoe and Janssens, 1988*; *Tyndale-Biscoe and Renfree, 1987*). During their early postnatal life marsupials attach to the mother's teat and use the comparatively early developed tongue musculature to suck (*Smith, 1994*). In the grey short-tailed opossum, *Monodelphis domestica*, pups are born after 13 days of embryonic development, which is followed by around 14 days permanently attached to the mother's teat, after which they detach intermittently from the mother but continue to suckle. Weaning occurs around postnatal day 60 (*Keyte and Smith, 2008*). In contrast to therian mammals, young monotremes do not obtain milk in quite the same way as therian mammals due to the absence of teats in the mother (*Griffiths, 1978*). Instead young monotremes suck up milk vigorously from the flattened but protuberant nipple-like areola on the mother's abdomen (*Griffiths, 1978*). In the case of echidnas, these areolae are within the pouch.

Given the lack of a jaw joint at birth, it has been proposed that marsupials and monotremes use the connection between the middle ear bones and cranial base to permit feeding prior to the formation of the articulation between the dentary and squamosal and cavitation of the middle ear (*Crompton and Parker, 1978*; *Maier, 1987*; *Sánchez-Villagra et al., 2002*; *Zeller, 1993*). To investigate this idea further, we have analysed the articulations that link the lower jaw to the cranial base (cranio-mandibular joints) in monotremes (platypus *Ornithorhyncus anatinus* and short-beaked echidna *Tachyglossus aculeatus*) as they develop from hatching, and compare them to a marsupial (grey short-tailed opossum, *Monodelphis domestica*), and a eutherian (mouse, *Mus musculus*), with additional comparison to the gecko, guinea pig and bat. We show that in early post-hatching life the monotreme incus and cranial base fuse, and later form an articulation, creating a double cranio-mandibular articulation, similar to the jaw anatomy of fossil mammal-like reptiles. This close association of the incus and cranial base is also observed at embryonic stages in eutherians and is reflected in mouse cell lineage studies. In contrast, opossums at birth utilise a cushion of extra-cellular matrix-rich mesenchyme in between the incus and petrosal to provide an articulation point. Marsupials and monotremes, therefore, have different strategies for coping with an early birth. Our research suggests that the incus retains a transient lower jaw support role across extant mammals but at different stages of pre and postnatal development.

## Results

### The primary jaw joint (malleus-incus) does not provide a site of articulation in marsupials and monotremes at birth

It has been suggested that the joint between the malleus and incus might act as the jaw joint early on in marsupial postnatal development, thereby recapitulating the reptilian function of these bones in mammals (*Müller, 1968*; *Crompton and Parker, 1978*). Alternatively, it has been suggested that the actual articulation point in marsupials is between the incus and the cranial base (*Maier, 1987*; *Sánchez-Villagra et al., 2002*). Less information is available regarding monotreme development, however, the incus has been described as being in cartilaginous connection with the cranial base during early post hatching development (*Watson, 1916*; *Zeller, 1993*). The development of the malleus and incus, and incus and cranial base, was therefore investigated across the three groups of mammals, with the gecko as an outgroup.

In the ocelot gecko (*Paroedura picta*), the quadrate and articular (the homologous elements to the incus and malleus respectively in non-mammal amniotes) form a clear synovial joint in the embryo at mid-gestation (*Figure 2A*). In mice (*Mus musculus*), the malleus and incus are initially formed from a single cartilaginous condensation that separates, by the formation of a joint, at Embryonic (E) day 15.5 (*Amin and Tucker, 2006*). At birth, therefore, the incus and malleus are evident as distinct

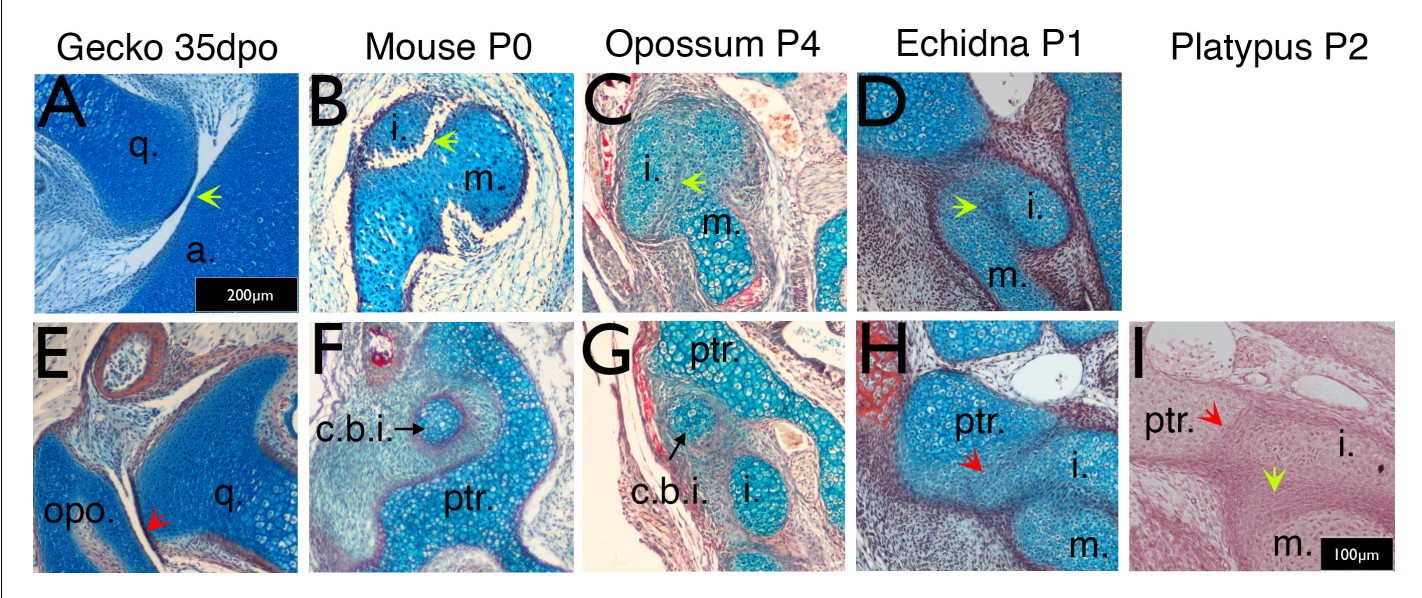

**Figure 2.** Timing of the development of the quadrate-articular/malleus incus, and cranio-incudo joints. Histological sections stained with alcian blue and picrosirius red. (A) The primarily jaw articulation is formed by 35 days of post-oviposition (35dpo) during in ovo development in geckos. (B) The malleus-incus joint, the homologue of the quadrate-articular joint, is formed during in utero development in mice, and is fully formed at birth (Postnatal day (P) 0). (C-D) The malleus incus joint is still partially fused in 4 day postnatal (P4) opossum pups (C) and 1 day post-hatching echidna young (P1) (D). (E) During development the gecko quadrate forms a joint with the opisthotic (structurally equivalent to the mammalian petrosal). (F) At birth there is no articulation between the crus breve of the incus and the surrounding crista parotica of the petrosal in mice (P0). (G) The crus breve of the incus sits in close proximity to the petrosal in opossums at P4 (G). (H-I) The incus is fused with the petrosal in both P1 echidna (H) and the P2 platypus (I). Green arrows highlight Q-A/M-I interaction. Red arrows highlight Incus/Q-petrosal/opisthotic interaction. a. articular; c.b.i crus breve of the incus; (i) incus; m. malleus; opo, opisthotic; ptr. petrosal; q. quadrate. Scale in A = 200 microns, same scale in E. Scale bar in I = 100 microns, same scale in B-D, F-H. The online version of this article includes the following figure supplement(s) for figure 2:

**Figure supplement 1.** 3D Reconstruction of cartilaginous middle ear ossicles and cranial base from histological sections shows differences in anatomy in different groups of mammals during development.

cartilages (*Figure 2B*). In *Monodelphis domestica*, the malleus and incus are still connected at birth at the dorsal end by a ridge of cartilage (*Filan, 1991*; *Figure 2C*). We observed a similar connection between the malleus and incus in the echidna (*Tachyglossus aculeatus*) just after birth. Like the opossum, the middle ear ossicles were fused dorsally, indicating that they function as a unit (*Figure 2D*). These findings demonstrate that, like opossums, monotremes do not use the primary jaw joint as the craniomandibular articulation before the development of the dentary-squamosal joint.

We therefore investigated the relationship between the incus and the petrosal in the cranial base in mice, opossums, platypus and echidna, comparing the interaction to the developing joint between the quadrate and opisthotic in embryonic geckos. In many reptiles, as shown in the gecko, the quadrate (incus homologue) forms a synovial joint with the opisthotic (also known as the otoccipital) in the cranial base during embryonic development (*Figure 2E*). The opisthotic/otoccipital is architecturally equivalent to the petrosal of mammals. In mice, the crus breve (short process) of the incus nestled in a fossa created by the crista parotica of the petrosal, but was separated by a region of mesenchymal cells, highlighting the lack of a clear articulation point between the two elements (*Figure 2F*). The incus at birth, therefore only articulated with the adjacent middle ear bones, the malleus and stapes. Similar to the mouse, the crus breve in neonatal opossums, fitted into a fossa created by the crista parotica, but abutted the petrosal on the inferior aspect of the crista parotica (*Figure 2G*). The incus and petrosal were therefore positioned much closer than in the mouse.

The relationship between the incus and crista parotica in the two monotreme species was significantly different from the therian mammals. In both platypus (*Ornithorhynchus anatinus*) and echidna (*Tachyglossus aculeatus*), the incus appeared to be fused with the crista parotica at birth (*Figure 2H, I*), agreeing with *Watson, 1916*. The lower jaw, via Meckel's cartilage, would therefore be physically

connected to the upper jaw, via the incus at this timepoint. 3D reconstructions of the incus, malleus and petrosal, showing the relationship of these different elements in the different species is shown in *Figure 2—figure supplement 1*. The relatively small size of the incus in both monotremes is striking, as is the extended and tapered crus breve of the incus in the opossum.

## Development of an incus-petrosal joint in monotremes during early feeding

To investigate the monotreme relationship between the incus and crista parotica further we followed development of these two cartilages from birth to functional use of the dentary-squamosal joint, but before complete cavitation of the middle ear space. Due to the scarcity of available specimens very little is known about monotreme ear and jaw development. In adult platypuses, the incus appears in contact with the crista parotica, forming a fibrous articulation (*Zeller, 1993*; *Luo and Crompton, 1994*). Similarly, in the adult echidna, the incus has been described as tightly attached to the petrosal (*Aitkin and Johnstone, 1972*).

At 2 days and 6.5 days the platypus incus was fused to the crista parotica by immature chondrocytes (*Figure 3A,B*). Between 10 days and 30 days the connection was difficult to make out, with the two cartilages almost completely integrated together (*Figure 3C,D*). Strikingly, by 80 days, when the dentary-squamosal joint would have started to become functional, the incus and crista parotica were no longer fused, with the two distinct cartilages abutting each other (*Figure 3E*). At this stage, in contrast to the other stages investigated, the ear ossicles and petrosal had begun to ossify. However, the regions forming the malleus-incus joint, and the incus-petrosal articulation remained cartilaginous. A cartilaginous articular surface between the incus and petrosal was maintained at 120 days, a period when the young would have started to leave the burrow (*Figure 3F*; *Holland and Jackson, 2002*). A similar move from early fusion, to articulation was observed in the echidna (*Figure 3F–J*). No evidence of a synovial capsule, however, was identified at any stage.

The fusion of the incus and crista parotica coincides with the period when the young would have been feeding from milk, while the move to an articulation was associated with periods when the dentary-squamosal was fully formed and functional. After separation of the incus and petrosal, there was

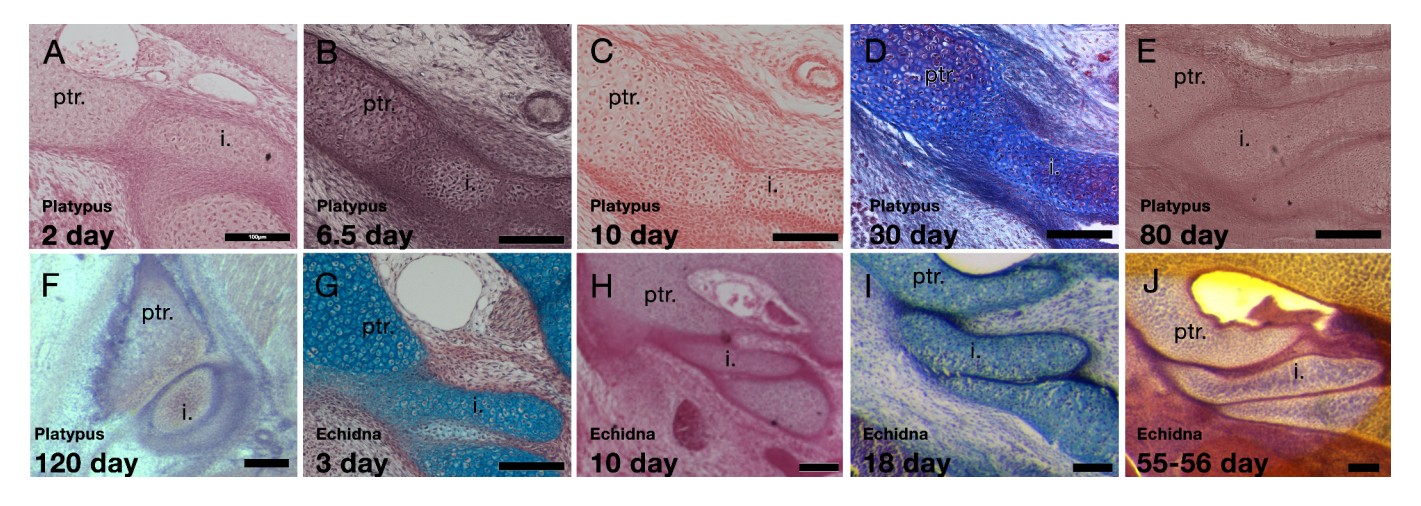

**Figure 3.** Development of the incus-petrosal joint in monotremes. (A-B) The platypus incus is fused to the petrosal by immature chondrocytes at 2 days (A) and 6.5 days (B) post-hatching. (C) At 10 days post-hatching, the fusion persists, with mature chondrocytes forming the connection. (D) A similar morphology is seen at 30 post-hatching. (E) At 80 days post-hatching the incus and petrosal are no longer fused, but instead the two cartilages abut each other. (F) At 120 days post-hatching the incus and petrosal have begun to ossify, but the region of articulation in between the two elements remains cartilaginous. (G-H) In echidna the incus is fused to the petrosal by immature chondrocytes at 3 days (G) and 10 days (H) post-hatching. (I-J) By 18 days post-hatching the two elements are separated but remain abutted (I), This connection remains though to 55–65 days post-hatching (J). i: incus; ptr. petrosal. Scale bar = 100 microns.

The online version of this article includes the following figure supplement(s) for figure 3:

**Figure supplement 1.** The middle ear and jaw joints of a 50 day old platypus.

a period where two cranial-mandible articulations were evident in the platypus - between Meckel's cartilage and the petrosal, via the malleus and incus, and between the dentary and squamosal (*Figure 3—figure supplement 1C*). The chain of elements linking Meckel's cartilage to the petrosal in the platypus is shown in *Figure 3—figure supplement 1A, B, D* at 50 days post-hatching.

Middle ear cavitation occurred very late in the monotreme specimens analysed, with only the 120 day platypus showing partial cavitation around the hypotympanum, but this did not extend upwards to where the ossicles are housed. Hearing, thus, must be a very late developing sense in the platypus.

## Upregulation of Wnt signalling initiates joint formation between the ossicles and cranial base in echidna

Limited expression analysis has been performed in monotremes, with no previous expression data performed in the ear or jaw during development. In order to further understand the change in the relationship between the incus and petrosal, immunohistochemistry staining was carried out in echidna samples 0 and 3 days post hatching.

In the fused incus-petrosal region of 0-day-old echidna (*Figure 4A*), the expression of both a master regulator of cartilage development, Sox9, and a principal component of cartilage extra cellular matrix, Collagen Type 2, were continuous between the incus and the crista parotica of the petrosal, as well as between the incus and the malleus (*Figure 4B*). Since the connection between these elements is lost later in post-hatching development, IF for beta-catenin was carried-out. Nuclear localised beta-catenin is a readout of canonical Wnt signalling, and is known to negatively regulate chondrocytes differentiation and promote joint formation (*Hartmann and Tabin, 2001*). Few beta-

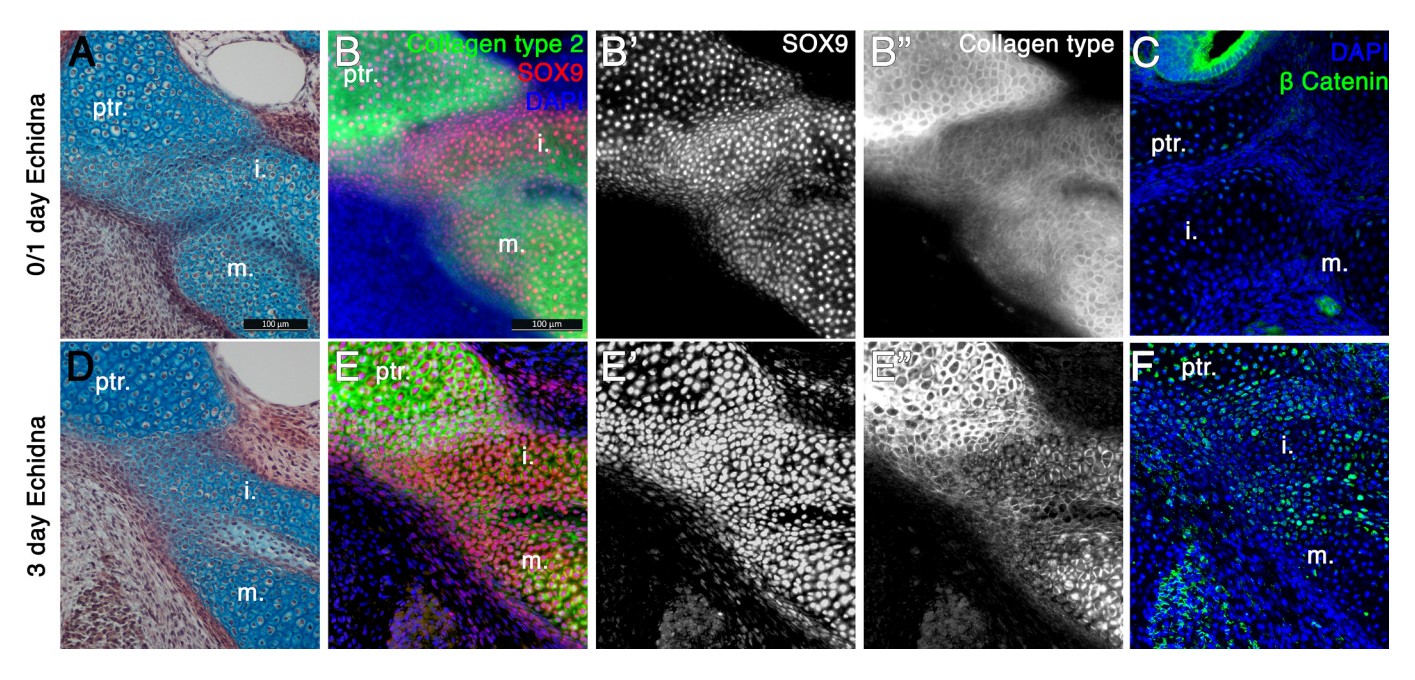

**Figure 4.** Fusion of the Incus with the petrosal in Echidna pouch young. (A) Alcian blue/picrosirius red staining on the fusion between the incus and petrosal observed in the newly hatched echidna. (B) Immunofluorescence staining against the regulator of chondrogenesis Sox9 (red) (B,B') and the marker of mature cartilage Collagen type 2 (green) (B,B") demonstrates that the cartilaginous incus and petrosal bones are fully fused at post-hatching day 0/1 (P0/1). (C) Immunohfluorescence against β Catenin (green) shows no activity within the cartilages at this timepoint. Expression is observed in the neuroepithelium of the inner ear. (D) Alcian blue/picrosirius red staining on the fusion between the incus and petrosal observed in 3 day post-hatching echidna (P3) shows that the elements are now fused by fibrocartilage. (E) Immunofluorescence staining against the regulator of chondrogenesis Sox9 and the marker of mature cartilage collagen type 2 (E,E"). Sox9 is still continuously expressed between the elements (E,E'), but collagen type 2 is down regulated in the incus-petrosal and incus-malleus articulation region (E,E"). (F) Immunofluorescence against β Catenin shows nuclear localisation within the incus-petrosal and incus-malleus articulation regions, indicating active canonical Wnt signalling, an important step in suppression of chondrogenesis during joint formation. i. incus; m. malleus; ptr. petrosal.

catenin positive cells were observed within the cartilage of the middle ear and petrosal at 0 days, though beta-catenin was strongly expressed in the neuro-epithelium of the inner ear (*Figure 4C*). At post-hatching day 3, the incus and crista parotica were still fused, although the cells joining the two elements resembled fibrocartilage or immature chondrocytes (*Figure 4D*). Expression of Sox9 was still strong and continuous throughout all elements (*Figure 4E,E'*), however Collagen Type 2 expression was weaker in the fusion region (*Figure 4E,E''*), possibly indicating a change in cartilage type from hyaline cartilage to fibrocartilage. Interestingly nuclear beta-catenin, suggestive of active Wnt signalling, was observed in two stripes, in the chondrocytes between the incus and petrosal, and within the malleus-incus joint, indicating suppression of cartilage fate in these regions (*Figure 4F*). Upregulation of Wnt signalling between the incus and petrosal therefore, may play a role in formation of a joint between these two, initially fused, structures.

## Interactions between the petrosal and incus are also observed prenatally in eutherian mammals

While the fusion between the incus and petrosal in echidna and platypus could be explained by the evolutionary distance between monotremes and therian mammals, it has also been suggested that the incus is transiently attached to the cranial base in 7-week-old human fetuses (*Rodríguez-Vázquez et al., 2018*). This suggests that the potential for fusion may be a default state in mammals. In order to examine this, we next undertook fate mapping experiments in the mouse, and investigated the relationship between the incus and petrosal in other eutherian mammals during embryonic development.

Sox9 expressing cells were fate mapped by tamoxifen induction at E14.5 in *Sox9CreERT2; tdTomato* mice, which were then collected at P0 (*Figure 5A*). At this stage Sox9 (green) was expressed in the petrosal and incus and suspensory ligaments, overlapping with the red fluorescent Protein (RFP) marking the Sox9 lineage cells. In addition, the red Sox9 lineage cells were found in the Sox9 negative mesenchymal cells, in the gap between the petrosal and incus (*Figure 5A*). A pre-cartilaginous bridge is therefore evident in the mouse between the incus and the crista parotica. Next, expression of Sox9 was investigated at E14.5. The incus, and the crista parotica are both neural crest derived (*O'Gorman, 2005*; *Thompson et al., 2012*), while the rest of the petrosal is mesodermal. We therefore looked at the expression of Sox9 (red) in *Mesp1Cre;mTmG* mice, where mesoderm-derived tissue can be detected by anti-GFP IF (*Figure 5B*). Since tissue processing and wax embedding removes endogenous fluorescence, the membrane RFP that is expressed in the non-mesodermal tissue of *Mesp1Cre;mTmG* mice was not detectable in these slides. Consequently, all red signal was Sox9 immunofluorescence staining. Sox9 protein was expressed continuously between the incus and the petrosal. The incus Sox9 expression domain was continuous with the expression domain of the neural crest -derived crista parotica, which in turn was fused to the mesodermal portion of the petrosal. Since the incus does not fuse with the petrosal in the mouse, despite the expression of Sox9 between the elements, we next looked at the mRNA expression of joint markers *Gdf5* and *Bapx1* between the incus and petrosal of mice by in situ hybridisation (*Figure 5C–E*; *Storm and Kingsley, 1999*; *Francis-West et al., 1999*; *Tucker et al., 2004*). *Gdf5* was expressed in the mesenchyme between the incus and petrosal, as well as in the malleus-incus joint (*Figure 5D*). *Bapx1*, which specifies both the malleus-incus joint and the quadrate-articular joint (*Tucker et al., 2004*), was not expressed in between the incus and the petrosal (*Figure 5E*). In the mouse, therefore there is a potential for the incus and crista parotica to fuse but they are prevented from doing so by the upregulation of the joint marker Gdf5.

Very close associations between the incus and crista parotica during development were also observed in other eutherian mammals via PTA stained microCT (see bat in *Figure 5—figure supplement 1*), suggesting that interactions between these two elements are observed as a feature prenatally in eutherian mammals, similar to post-hatching monotremes. The function of this prenatal connection between the upper and lower jaw is unclear but may act as a brace to buffer movement during this period.

## Petrosal-incus relationships in marsupials

Next we investigated the articulation between the incus and petrosal observed in the developing opossum. It was originally suggested that the marsupial incus forms a joint with the crista parotica

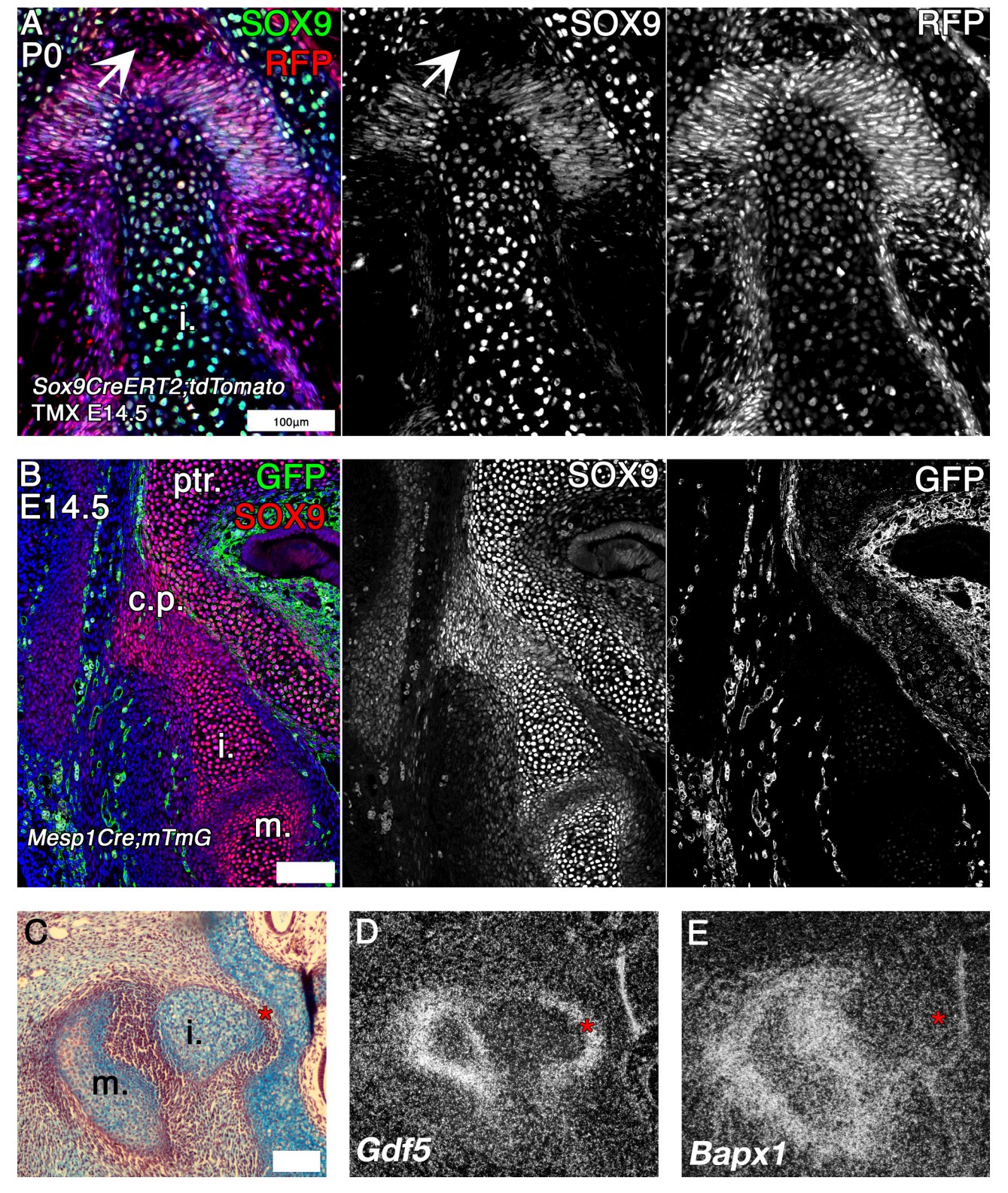

**Figure 5.** Mouse fate mapping studies demonstrate developmental fusion between incus and petrosal. (**A**) Genetic tracing of chondrogenic Sox9 expression cells by inducible reporter mice at postnatal day 0 (P0). Sox9 lineage cells (red) (RFP) are observed in the mesenchyme and developing ligaments between the crus breve of the incus and the petrosal. Sox9 protein (green) is not expressed in the mesenchyme surrounding the incus at P0 (arrowhead). (**B**) Genetic tracing of mesoderm lineage cells (green) (GFP) and immunohistochemistry against Sox9 protein (red) at embryonic day 14.5

*Figure 5 continued on next page*

Figure 5 continued

(E14.5). Sox9 expression at E14.5 confirms that the incus and petrosal are formed from a continuous chondrogenic mesenchyme, and that the incus joins with the petrosal at the crista parotica, which is not of mesodermal origin. (C-E) Expression by in situ hybridisation of joint markers in sagittal section of E14.5 mouse middle ears. Gdf5 mRNA is expressed with the malleus-incus joint, and between the incus and the petrosal (D), potentially acting to inhibit the Sox9 expressing mesenchyme between the ear and the cranial base from differentiating into cartilage. The middle ear joint marker Bapx1 is not expressed between the incus and the petrosal (E). * indicates space between of incus and petrosal in C-E. i. incus; m. malleus; ptr. petrosal. Scale bar in A,B = 100 microns.

The online version of this article includes the following figure supplement(s) for figure 5:

**Figure supplement 1.** Contrast enhanced µCT of embryonic bat (Pterobnotus quadridens) middle ear at Carnegie Stage21, showing abutment of the crus breve of the incus against the petrosal.

(*Maier, 1987*), although this was disputed in *Monodelphis* (*Filan, 1991*). Although this latter paper found no evidence of a joint they did show the mesenchyme between the crista parotica and incus as being condensed (*Filan, 1991*). We therefore investigated the extra cellular matrix (ECM) components of the mesenchyme surrounding the opossum incus in more detail (*Figure 6*). It was noted that mesenchyme surrounding the crus breve and superior portion of the body of the incus had a more intense staining with alcian blue compared to those regions around the inferior border of the incus and the other ossicles (*Figure 2C,G*). This pattern was observed throughout ossicle development (*Figure 6A–C*). In order to further characterise the differences in the ECM in the different regions of the middle ear mesenchyme, immunohistochemistry for versican was carried out. Versican is a large proteoglycan with side chains of glycosaminoglycans (GAGs), such as hyaluronic acid (HA). Proteoglycan complexes act to attract water, and are held in place by collagen fibres to stiffen the matrix in hyaline cartilage, and act to lubricate articular cartilage (*Wu et al., 2005*). Versican is required during the initial condensation of mesenchyme but is absent from mature cartilage, where aggrecan is expressed (*Kamiya et al., 2006*). Versican expression is maintained in the joint region during limb cartilage development, acting to inhibit maturation of the mesenchyme to form cartilage (*Choocheep et al., 2010*; *Snow et al., 2005*).

Versican was strongly expressed in the mesenchyme surrounding the short arm of the incus at 5 days, 10 day and 27 days, correlating with the region of strong alcian blue expression (*Figure 6D–F*). The high level of versican around the crus breve therefore suggests a role for the ECM in providing a buffering function in this region. Cell density of the mesenchyme was measured in regions with strong alcian blue/versican staining and compared against the cell density of regions with low alcian blue/versican staining. Unpaired two-tailed t-test demonstrated that the regions with high alcian blue had a significantly higher (p=0.0152) cell density than those regions with lower alcian staining (*Figure 6G*).

Versican is processed by ADAMTS family members for clearing and remodelling (*Nandadasa et al., 2014*). While the full-length form of versican is thought to have a structural role, the cleaved form has an active role in signalling, influencing morphogenesis and tissue remodelling (*Nandadasa et al., 2014*). Interestingly when we analysed the cleaved form of versican, using antibodies against DPEAAE, the expression was largely reciprocal to that of uncleaved versican, with lower levels specifically around the crus breve (*Figure 6—figure supplement 1A*). This suggests that versican around the incus is protected from cleavage allowing it to maintain its structural role. The lack of cleaved versican around the crus breve, suggests the lack of a signalling role in this region, in agreement with the low level of expression of CD44, a cell surface receptor and binding partner of versican-hyaluronan complexes. CD44 was not associated with the mesenchyme around the crus breve, but was instead restricted to the perichondrium of the cartilaginous elements and periosteum of the skeletal elements of the ear (*Figure 6—figure supplement 1B*).

## Discussion

### Ear ossicles as transient jaw support in mammals

The incus of adult mammals plays a key role in hearing. Our data here suggest that the incus also plays a transient role supporting the lower jaw against the cranial base during both marsupial and monotreme postnatal development. The role of the incus and the points of jaw articulation are

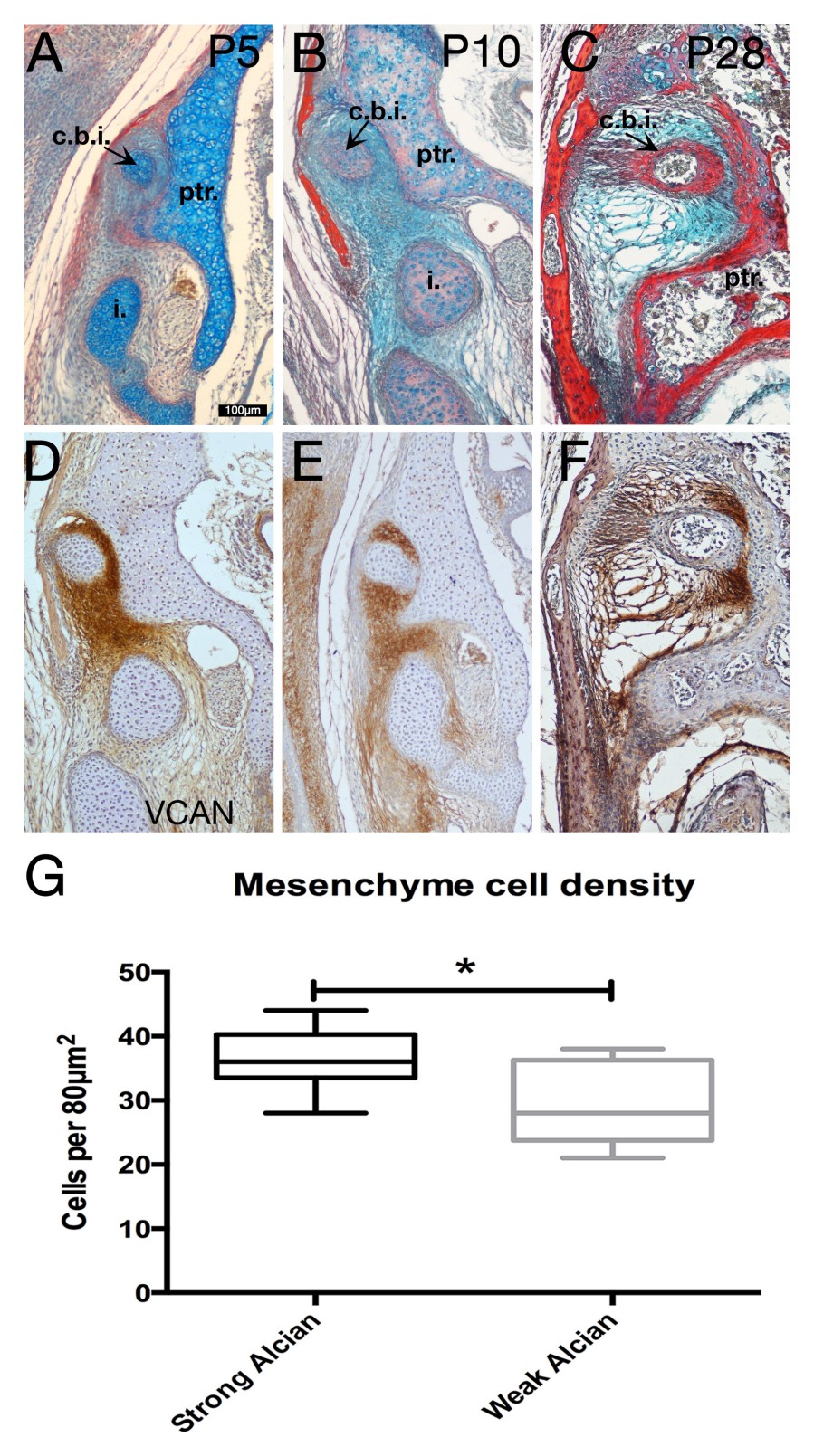

**Figure 6.** Specialist mesenchyme supports incus-petrosal connection in juvenile opossums. (A-F) Mesenchyme surrounding the crus breve of the incus is rich in the proteoglycan Versican (Vcan) at postnatal day (P)5 (A,D) and P10 (B,E). During cavitation of the middle ear at P28 versican rich mesenchyme is concentrated between the crus breve of the incus and the petrosal (C,F). (G) At P5 the proteoglycan-rich regions surrounding the crus breve have a

*Figure 6 continued on next page*

*Figure 6 continued*

significantly greater cell density than the regions with less proteoglycan. *p=0.0152 unpaired two-tailed t-test. Error bars = 1 standard deviation. c.b.i crus breve of the incus; i. incus; ptr. petrosal. Scale bar in A = 100 microns, same scale in **B-F**.

The online version of this article includes the following source data and figure supplement(s) for figure 6:

**Source data 1.** Statistical analysis of cell number in specialised mesenchyme surrounding the opossum incus.

**Figure supplement 1.** Versican signaling is not evident in the mesenchyme around the opossum incus.

summarised in *Figure 7A*. The incus and petrosal were found to be fused at hatching in both monotremes. During this early fusion period, the puggle would be feeding exclusively on milk and Meckel's cartilage could therefore act as a flexible elastic strut to facilitate jaw movement (*Zeller, 1993*).

Interestingly, a potential role of the ear ossicles in jaw support was also observed in eutherians during prenatal development. Fate mapping and gene expression studies in mice indicated that the crus breve of the incus and the crista parotica were formed from a continuous region of Sox9 expressing chondrogenic cells (*Figure 5A, B*), separated by expression of the joint marker *Gdf5* (*Storm and Kingsley, 1999*; *Figure 5C*). Furthermore, the incus and cranial base temporarily fuse during the development of the human middle ear region (*Rodríguez-Vázquez et al., 2018*), and abut during bat development (*Figure 5—figure supplement 1*). Together these data indicate that the relationship of the incus to the cranial base is not a derived feature of monotremes, and that the common mammal-like reptile ancestors of both monotremes and therian mammals may have formed an articulation between the quadrate/incus and petrosal through fusion of the elements followed by joint formation though Wnt and Gdf5 signalling.

The current study indicates that the first pharyngeal arch-derived incus forms a continuous field of chondrocytes with the second arch-derived crista parotica, which in turn is fused with the mesoderm-derived body of the petrosal. The borders between these developmentally distinct populations are, therefore, not always reflected by the mature anatomy.

For young monotremes and marsupials, the middle ear must function as part of the mandible postnatally until the dentary-squamosal bones have formed. This is similar, but not identical to the situation in cynodont ancestors of mammals. In these animals, the quadrate/incus articulated with a number of cranial elements, including the quadratojugal, to stabilise the jaw articulation. These connections and many elements like the quadratojugal have been lost in extant mammals in order to free the incus and increase its mobility during sound transmission (*Luo and Crompton, 1994*). The mechanical requirements for feeding placed upon the middle ears in monotremes and marsupials during early life have resulted in the fusion of the incus and petrosal in monotremes, and the elongated contact supported by a proteoglycan matrix in marsupials. These adaptations allow for stabilisation of the middle ear before the development of the dentary-squamosal joint and separation of the middle ear from the mandible, but do not compromise the effectiveness of the middle ear in later life. The changing connections between the middle ear ossicles and the cranial base in the different groups are highlighted in *Figure 7B*.

## The opossum has a specialised anatomy to brace the middle ear against the cranium during sucking

The crus breve of the incus is elongated in the developing opossum compared with other species analysed (*Figure 2—figure supplement 1*). In order to feed by suckling in the absence of a dentary-squamosal joint we propose that this anatomy allows for an increased surface contact with the cranial base during postnatal development, which, in combination with the proteoglycan-rich surrounding mesenchyme, acts to stabilise the mandible against the rest of the head. It is noted that many adult marsupials have a relatively elongated crus breve of the incus compared to eutherian species, for example the bare-tailed woolly opossum *Caluromys philander,* and the grey short-tailed opossum, *Monodelphis domestica* (*Sánchez-Villagra et al., 2002*). Even when eutherian mammals have a longer crus breve, such as in Talpid moles, the process is thinner and more finger-like compared to that of marsupials (*Segall, 1973*; *Segall, 1970*). This may be a consequence of the developmental requirement for an elongated short process to facilitate feeding before the development of the mature mammalian jaw articulation.

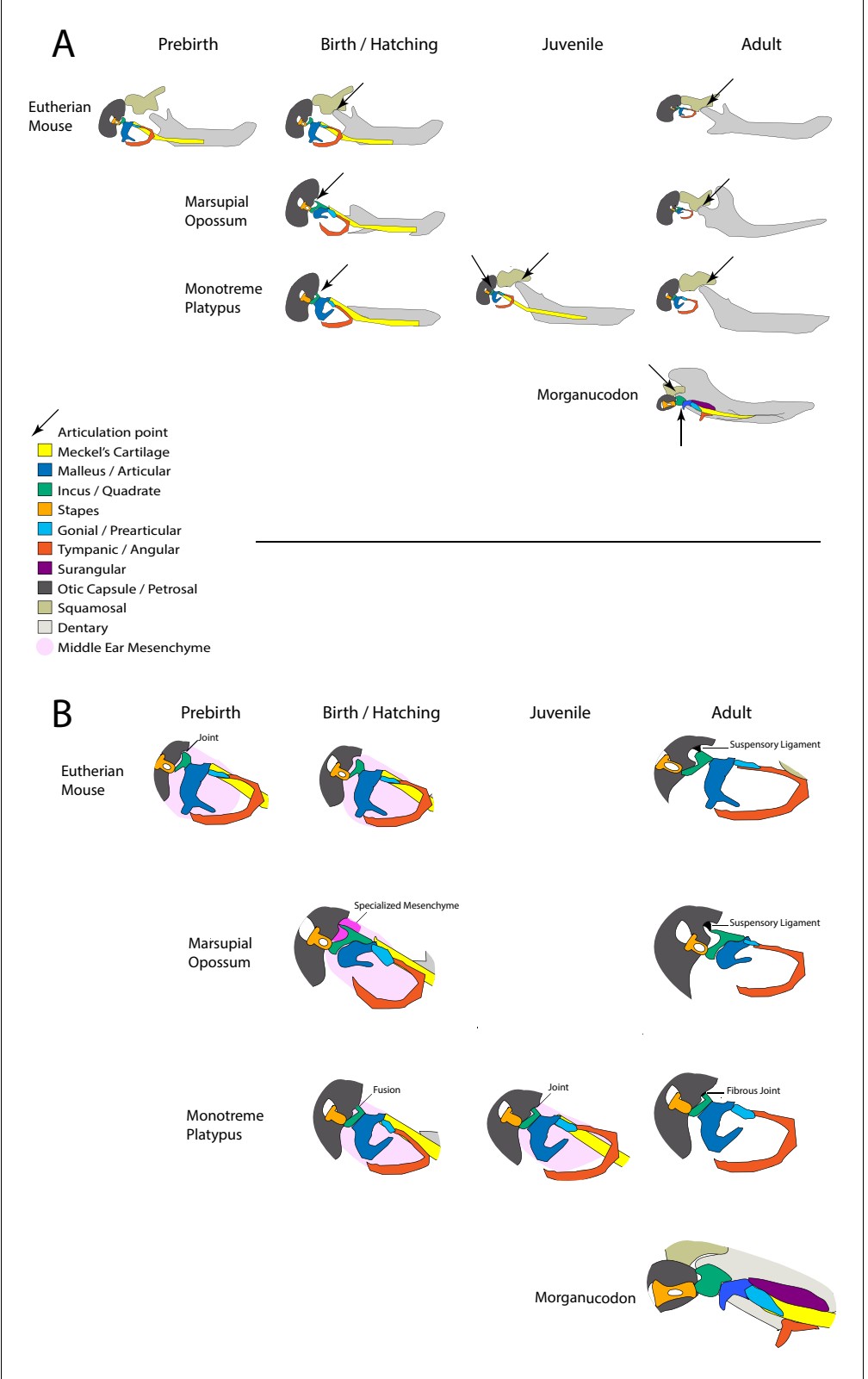

**Figure 7.** Summary of involvement of middle-ear ossicles in jaw articulation during development. (**A**) The location of the jaw articulation in developing living mammals and in the extinct mammal-like reptile *Morganucodon*. Arrows indicate jaw articulation points. Eutherian mammals are born with a functional dentary-squamosal joint (TMJ), while young marsupials and monotremes use the middle ear bones due to a lack of this joint, which develops later. During postnatal development monotremes show evidence of a double jaw articulation, similar to fossil mammal-like reptiles such as

*Figure 7 continued on next page*

*Figure 7 continued*

*Morganucodon*. (B) The connections between the middle ear ossicles and the cranial base in developing mammals. The connections between the incus and cranial base differ in young marsupials and monotremes. The fusion followed by a joint seen in monotremes is also observed in pre-natal eutherians. Neonatal marsupials support the incus with a specialised middle-ear mesenchyme.

In the majority of adult marsupials, including *Monodelphis*, the incus is suspended from the cranial base by suspensory ligaments, and the crus breve extends into a fossa. One interesting exception is the marsupial mole, in which the crus breve has a connective tissue attachment to a lamella on the petrosal (*Archer, 1976*). This results in the middle ear ossicles being affixed to the cranial base, an adaptation to a fossorial niche found in other mammals such as in true moles. In light of the current study, the absence of an incudal fossa in the marsupial mole may be interpreted as a retention of the juvenile petrosal morphology (paedomorphy).

## Consequence of ECM in opossum middle ear

In adult non-mammalian amniotes the homologue of the incus - the quadrate - and cranial base are strongly attached by fibrous syndesmoses or cartilaginous synchondroses (*Payne et al., 2011*), and we show that a synovial joint appears to form in geckos during development (*Figure 2*). In the neonatal opossum neither type of connection is observed. In neonatal marsupials Sánchez-Villagra and colleagues describe the connection between the incus and petrosal as being an 'immature syndesmosis', which acts as a 'supportive strut' during sucking (*Sánchez-Villagra et al., 2002*). In the current study, we demonstrate a specialised condensed mesenchyme surrounds the incus of opossum postnatal juveniles. We show that this condensed mesenchyme is rich in the proteoglycan versican (*Figure 6*). In contrast expression studies in human foetuses demonstrate that versican is restricted to the perichondrium of Meckel's cartilage (*Shibata et al., 2014*; *Shibata et al., 2013*), with high hyaluronic acid levels within the joints but not surrounding the incus (*Takanashi et al., 2013*). This concentration of versican around the crus breve therefore may be a feature of *Monodelphis*, and perhaps marsupials in general.

The versican-rich mesenchyme may act to either stabilise the incus by increasing the tension of the surrounding mesenchyme during feeding, 'lubricate' the articulation between the incus and cranial base by increasing the hydration of the ECM, or both. In keeping with this role, versican is dynamically expressed at the pubic symphysis during pregnancy in mice (*Rosa et al., 2012*), during which time the mouse pubic symphysis forms a fibrous joint or syndesmosis (*Ortega et al., 2003*). Significantly, there is little cleaved versican (DPEAAE) around the crus breve of the incus, suggesting a mechanical, rather than a signalling role (*Figure 6—figure supplement 1. A*). Overall it is likely that this mesenchyme is supporting the incus, rather than enabling mobilisation, with the high level of uncleaved versican acting to increase fibroviscocity while also elevating hydration of the ECM. In this way, the mesenchyme around the incus acts as a cushion during the mechanical stress of suckling.

## A double jaw articulation during monotreme development

Meckel's cartilage persists to at least 50 days post-hatching in the platypus. At this timepoint, juvenile monotremes have two connections between the lower and upper jaw. The first connection is through the middle ear, which in juveniles remains attached to the mandible and articulates with the cranial base via the incus. The second is the later developing novel mammalian jaw joint. Only much later in the life of the young does it appear that the connection between the middle ear and mandible is lost, and the malleus and incus act as a DMME. The connection of the incus to the cranial base appears to be maintained in the adult echidna and platypus (*Luo and Crompton, 1994*; *Aitkin and Johnstone, 1972*). This would be expected to impact on the movement of the incus, and therefore the efficiency of hearing, reflected in the poor hearing reported for monotremes (*Aitkin and Johnstone, 1972*; *Gates et al., 1974*).

This novel finding of a double cranial articulation in the juvenile has significant implications for the evolution of the middle ear and jaw joint in mammals. Fossil evidence indicates that mammalian ancestors had a persistent connection between the middle ear ossicles and the jaw, as evidenced by the presence of an ossified Meckel's element, or a dentary groove and post dentary trough,

supporting a persistent Meckel's cartilage (*Luo, 2011*; *Rich et al., 2005*; *Urban et al., 2017*). For these animals, the connection of the middle ear with the jaw took one of two forms, in each case the mammalian secondary jaw joint was present. The first was a more basal mandibular middle ear where the incus and malleus were firmly attached to the cranial base and dentary respectively. More derived fossils had a partial, or transitional mammalian middle ear (PMME or TMME), where the middle ear was medially inflected away from the dentary, presumably allowing for improved vibration, but the malleus was still connected to the jaw, via Meckel's cartilage (*Luo, 2011*). In these fossils with a PMME, little is understood of the rear of the ossicular chain, where the incus meets the petrosal, due to the poor and rare preservation of middle ear ossicles in the fossil record, a consequence of their small size. For example, only recently has a multituberculate with a complete incus been described (*Wang et al., 2019*). Our data suggest that even in these transitional mammals with a PMME, the incus would have still articulated with the cranial base via the crista parotica, at least at some point during the animal's life history.

The DMME appears to have evolved independently in monotremes and therian mammals (*Rich et al., 2005*). Due to the absence of evidence we do not know if the incus articulation in animals with a PMME varied in a lineage specific manner, with the therian lineage resembling juvenile marsupials, and monotremaformes resembling juvenile platypuses and echidna, or if both lineages had similar articulations. The data from transgenic reporter mice (*Figure 5*), along with data from humans (*Rodríguez-Vázquez et al., 2018*) and non-model therians (*Figure 5—figure supplement 1*) suggests that the monotreme-type fusion and articulation of the incus with the cranial base may have been common in mammal like-reptiles. As such, the developing monotreme, with a double jaw articulation and a fused or articulated incus and petrosal, provides an exciting model for the study of the developmental basis of mammalian evolution.

## Materials and methods

**Key resources table**

| Reagent type (species) or resource | Designation | Source or reference | Identifiers | Additional information |
|---|---|---|---|---|
| Strain, strain background (*Monodelphis domestica*) | short-tailed opossum | *Anthwal et al., 2017*. DOI: 10.1038/s41559-017-0093; *Urban et al., 2017*. DOI: 10.1098/rspb.2016.2416 | | |
| Strain, strain background (*Cavia porcellus*) | Guinea pig | *Anthwal et al., 2015*. DOI: 10.1186/s13227-015-0030-6 | | |
| Strain, strain background (*Paroedura picta*) | ocelot gecko | *Zahradnicek et al., 2012*. DOI: 10.1111/j.1469-7580.2012.01531.x | | |
| Strain, strain background (*Mus musculus*) | CD1 | King's College London | | |
| Strain, strain background (*Tachyglossus aculeatus*) | short-beaked echidna | University of Melbourne | | |
| Genetic reagent (*Mus musculus*) | *Sox9CreERT2: tdTomato* | Other | RRID:MGI:4947114 | Prof Robin Lovell-Badge, Francis Crick Institute |
| Genetic reagent (*Mus musculus*) | *Mesp1Cre; mTmG* | RIKEN | Mesp1$^{tm2(cre)Ysa}$::Gt(ROSA)26S or$^{tm4(ACTB-tdTomato,-EGFP)Luo}$; RRID:MGI:3702469 | |
| Biological sample (*Pterobnotus quadridens*) | sooty mustached bat µCT scan | Other | | Karen Sears, UCLA |

*Continued on next page*

*Continued*

| Reagent type (species) or resource | Designation | Source or reference | Identifiers | Additional information |
|---|---|---|---|---|
| Biological sample (*Ornithorhynchus anatinus*) | platypus histological slides | Hill Collection, Museum für Naturkunde, Leibniz Institute for Research on Evolution and Biodiversity, Berlin; Cambridge University Museum of Zoology; *Green, 1937*; *Presley and Steel, 1978*; *Watson, 1916* | Specimen W; M45; Specimen Delta; M038; HP; Specimen Beta; HX | Museum Samples |
| Biological sample (*Tachyglossus aculeatus*) | short-beaked echidna histological slides | Cambridge University Museum of Zoology; *Green, 1937*; *Presley and Steel, 1978*; *Watson, 1916*; | HX; Echidna H.SP EC5; Echidna H.SP EC4 | Museum Samples |
| Antibody | rabbit polyclonal anti Sox9 | Millipore | ab5535; RRID:AB_2239761 | IF 1/200 |
| Antibody | chicken polyclonal anti GFP | Abcam | ab13970; RRID:AB_300798 | IF 1/500 |
| Antibody | Rat monoclonal anti RFP | Chromotek | 5f8-100; RRID:AB_2336064 | IF 1/200 |
| Antibody | Rabbit polyclonal anti Beta-catenin | Santa Cruz | sc-7199; RRID:AB_634603 | IF 1/200 |
| Antibody | mouse monoclonal anti Tenascin C | DSHB | M1B4; RRID:AB_528488 | IF 1/40 |
| Antibody | mouse monoclonal anti type 2 collagen | DSHB | II-II6B3; RRID:AB_528165 | IF 1/50 |
| Antibody | mouse monoclonal anti CD44 | DSHB | HERMES-1; RRID:AB_528148 | IHC 1/50 |
| Antibody | mouse monoclonal anti Versican | DSHB | 12C5; RRID:AB_528503 | IHC 1/50 |
| Antibody | rabbit polyclonal anti Versican V1 (DPEAAE) | Abcam | ab19345; RRID:AB_444865 | IF 1/400 |
| Antibody | Donkey Polyclonal Alexa568 conjugated anti-Rabbit | Invitrogen | A10042; RRID:AB_2534017 | IF 1/300 |
| Antibody | Donkey Polyclonal Alexa 488 conjugated anti-Rabbit | Invitrogen | A-21206; RRID:AB_2535792 | IF 1/300 |
| Antibody | Donkey Polyclonal Alexa568 conjugated anti-Mouse | Invitrogen | A10037; RRID:AB_2534013 | IF 1/300 |
| Antibody | Goat Polyclonal Alexa488 conjugated anti-Chicken | Invitrogen | A-11039; RRID:AB_2534096 | IF 1/300 |
| Antibody | Goat Polyclonal Alexa568 conjugated anti-Rat | Invitrogen | A-11077; RRID:AB_2534121 | IF 1/300 |
| Antibody | Goat Polyclonal Biotin Anti-Mouse | Dako | E0433; RRID:AB_2687905 | IHC 1/400 |

*Continued on next page*

*Continued*

| Reagent type (species) or resource | Designation | Source or reference | Identifiers | Additional information |
|---|---|---|---|---|
| Recombinant DNA reagent | Mouse Gdf5 in situ hybridisation probe | *Tucker et al., 2004*. DOI: 10.1242/dev.01017 | | |
| Recombinant DNA reagent | Mouse Bapx1 in situ hybridisation probe | *Tucker et al., 2004*. DOI: 10.1242/dev.01017 | | |
| Commercial assay or kit | ABC-HRP streptavidin kit | Vector Labs | PK-6100; RRID:AB_2336819 | |
| Commercial assay or kit | ImmPACT DAB Peroxidase Substrate | Vector Labs | SK-4105; RRID:AB_2336520 | |
| Software, algorithm | FIJI | *Schindelin et al., 2015*; *Schindelin et al., 2012* | RRID:SCR_002285 | |
| Software, algorithm | Prism 8 | Graphpad | RRID:SCR_002798 | |
| Other | Aqueous mounting medium with DAPI | Abcam | ab104139 | |
| Other | Picro-sirius red, haematoxylin and alcian blue trichrome staining | CCRB Histology Core at King's College London | | |

## Animal tissues

Opossum (*Monodelphis domestica*) tissue was collected as previously described (*Anthwal et al., 2017*; *Urban et al., 2017*).

Archival platypus (*Ornithorhynchus anatinus*) and short-beaked echidna (*Tachyglossus aculeatus*) slides were imaged from the collections at the Cambridge University Museum of Zoology, and the Hill Collection, Museum für Naturkunde, Leibniz Institute for Research on Evolution and Biodiversity, Berlin. Details of samples imaged are in *Table 1*. All museum samples have been studied in previously published works (*Green, 1937*; *Presley and Steel, 1978*; *Watson, 1916*). Stages for platypus are estimated based on *Ashwell, 2012*. Staging of echidna H.SP EC5 and H.SP EC4 are estimated by cross-referencing (*Griffiths, 1978*; *Rismiller and McKelvey, 2003*). Post-hatching day 0 to 3 echidna samples were collected by Marilyn Renfree and Stephen Johnston.

Wildtype and *Mesp1Cre; mTmG* were kept at the King's College London Biological Services Unit. *Sox9CreERT2:tdTomato* embryos were a gift of Prof Robin Lovell-Badge and Dr Karine Rizzoti at the Francis Crick Institute, London.

Phosphotungstic acid (PTA) contrasted embryonic *Pterobnotus quadridens* bat μCT scans were provided by Prof Karen Sears and Dr Alexa Sadier at the University of California Los Angeles.

**Table 1.** Museum held monotreme specimens used in the current study.
CRL – Crown rump length. [*]Estimate based (*Ashwell, 2012*). [†]Estimate based on *Griffiths, 1978* and *Rismiller and McKelvey, 2003*.

| Species | Collection | ID | Estimated age | CRL/Max length |
|---|---|---|---|---|
| *Ornithorhynchus anatinus* | Cambridge | Specimen W | 2 days [*] | 16.5 mm |
| *Ornithorhynchus anatinus* | Berlin | M45 | 6.5 days [*] | 33 mm |
| *Ornithorhynchus anatinus* | Cambridge | Specimen Delta | 10 days [*] | 80 mm |
| *Ornithorhynchus anatinus* | Berlin | M038 | 30 days | Unknown |
| *Ornithorhynchus anatinus* | Cambridge | HP | 50 days [*] | 200 mm |
| *Ornithorhynchus anatinus* | Cambridge | Specimen Beta | 80 days [*] | 250 mm |
| *Ornithorhynchus anatinus* | Cambridge | HX | 120 days | 295 mm |
| *Tachyglossus aculeatus* | Cambridge | Echidna H.SP EC5 | 18 days [†] | 83 mm |
| *Tachyglossus aculeatus* | Cambridge | Echidna H.SP EC4 | 55–65 days [†] | 174 mm |

Guinea pig (*Cavia porcellus*) displays samples were collected as previously described (*Anthwal et al., 2015*).

Gecko and mouse samples were investigated during embryonic development (35 days post ovi-position (dpo) and E16.5 respectively). The gestation for geckos is around 60 days, and mice have a gestation of 20–21 days. Much of opossum and echidna development occurs during early post-gestation/hatching life, including formation of the dentary-squamosal joint, and so 4-day-old opossums, and 0- to 3-day-old echidnas were investigated before the onset of this joint.

All culling of mouse, opossum, guinea pig and reptile tissue followed Schedule One methods as approved by the UK Home Office and was performed by trained individuals. Use of genetically modified mice was approved by the local GMO committee at King's, under personal and project licences in accordance with the Animal (Scientific Procedures) Act of 1986, UK.

## Tissue processing and histological staining

All tissues for histological sectioning were fixed overnight at 4°C in 4% paraformaldehyde (PFA), before being dehydrated through a series of graded ethanol, cleared with Histoclear II, before wax infiltration with paraffin wax at 60°C. Wax-embedded samples were microtome sectioned at 8 µm thickness, then mounted in parallel series on charged slides.

For histological examination of bone and cartilage, the slides were then stained with picrosirius red and alcian blue trichrome stain using standard techniques.

## Immunofluorescence

For immunofluorescence staining slides were rehydrated through a graded series of ethanol to PBS. Heat induced antigen retrieval was carried out by microwaving the samples for 10 min in 0.1M Sodium citrate pH6 buffer. Slides were then blocked in 1% Bovine serum albumin, 0.1% cold water fish skin gelatine, 0.1% triton-X for 1 hr. Sections were then treated over night at 4°C with primary antibodies. The following primary antibodies were used, rabbit anti Sox9 (Chemicon) at a dilution of 1/200, chicken anti GFP (Abcam) at a dilution of 1/500, rat anti RFP (Chromotek) at a dilution of 1/200, Rabbit anti Beta-catenin (Santa Cruz) 1/200, mouse anti type 2 collagen (DSHB) at 1/50, mouse anti CD44 (DSHB) at 1/50, mouse anti Tenascin C (DSHB) at 1/40, mouse anti versican (DSHB) at 1/50, rabbit anti versican V1 (Abcam) at 1/400. Following repeated PBS washes, secondary antibodies were added. For fluorescent labelling the following antibodies were used at 1/300: Alexa568 conjugated Donkey anti-Rabbit, Alexa 488 conjugated Donkey anti-Rabbit, Alexa568 conjugated Donkey anti-Mouse, Alexa568 conjugated Donkey anti-Rat, Alexa488 conjugated Donkey anti-Chicken (all Invitrogen). Secondary antibodies were added in the blocking buffer for 1 hr at room temperature in the dark. The secondary antibody was then washed off with PBS, and the slides mounted with Fluroshield mounting medium containing DAPI (Abcam). Sections were visualised by Leica SP5 confocal microscopy. For Versican and CD44 slides, secondary biotinylated goat anti-mouse antibody (Dako) was added to the slides 1/400 in blocking buffer. Slides were then washed in PBS before being treated with ABC-HRP streptavidin kit (Vector Labs), and then revealed with DAB (Vector Labs).

Monotreme immunofluorescence staining was carried out in technical replicates due to the rare nature of the samples. Mouse and opossum analysis was carried out in biological triplicates.

## In situ hybridisation

Radioactively labelled antisense RNA probes were made against mouse *Gdf5* and *Bapx1* mRNA, and radioactive in situ hybridisations were carried out to detect the expression of these genes in sagittal plain cut sections of wildtype mice, as previously described (*Tucker et al., 2004*). All in situ staining was carried out in biological replicates.

## 3D reconstruction

Three-dimensional reconstructions of middle ear and surrounding cranial base cartilages were generated from serial histology images in FIJI (ImageJ 1.47 v), using the Trackem2 Plugin (*Schindelin et al., 2015*; *Schindelin et al., 2012*).

## Cell density counts

Cell density was counted in a DAPI stained sections of 5-day-old opossums (n = 3). In FIJI, 20 separate 80 µm$^2$ fields were randomly placed across the mesenchyme surrounding the incus across 5 sections. The total number of nuclei were counted if they were located wholly within the field, or where more than 50% of the nuclei intersected the upper or right hand margin of the field. Next by looking at parallel alcian blue stained sections, the fields were scored as being in proteoglycan-rich or weak regions. Two fields were ambiguous, and so were removed from the analysis. The user did not know the proteoglycan status of the field at the time of counting. Next the mean cell number in each field was calculated in the remaining 8 proteoglycan-rich (alcian blue stained) regions and 10 proteoglycan weak (weak alcian blue stain) regions, and compared by unpaired two-tailed students t-test in Prism statistical analysis software (Graphpad), with $p < 0.05$.

## Acknowledgements

Thanks to Karen Sears (UCLA) for the bat microCT scan. Thanks to Robert Asher for access to monotreme samples held at the Zoological Museum in Cambridge University, and to Andrew Gillis for imaging. Thanks to Peter Giere for access to the Hill Collection at the Berlin Museum fur Naturkunde. Thanks to Prof Robin Lovell-Badge and Dr Karine Rizzoti at the Francis Crick Institute, London for provision of the *Sox9cre/Tom* mice. Mesp1cre mice were provided through an MTA agreement with RIKEN, Japan. Thanks to Currumbin Wildlife Sanctuary for the echidna pouch young samples.

## Additional information

### Funding

| Funder | Grant reference number | Author |
|---|---|---|
| Wellcome | 102889/Z/13/Z | Abigail S Tucker |
| Australian Research Council | Linkage Grant | Stephen D Johnston<br>Marilyn B Renfree |

The funders had no role in study design, data collection and interpretation, or the decision to submit the work for publication.

### Author contributions

Neal Anthwal, Conceptualization, Formal analysis, Investigation, Methodology, Writing - original draft, Writing - review and editing; Jane C Fenelon, Stephen D Johnston, Marilyn B Renfree, Resources, Writing - review and editing; Abigail S Tucker, Conceptualization, Supervision, Funding acquisition, Investigation, Writing - review and editing

### Author ORCIDs

Neal Anthwal (iD) https://orcid.org/0000-0002-4104-7839
Jane C Fenelon (iD) https://orcid.org/0000-0001-8771-5196
Marilyn B Renfree (iD) https://orcid.org/0000-0002-4589-0436
Abigail S Tucker (iD) https://orcid.org/0000-0001-8871-6094

### Ethics

Animal experimentation: This study was performed under UK Home Office licence and regulations in line with the regulations set out under the United Kingdom Animals (Scientific Procedures) Act 1986 and the European Union Directive 2010/63/EU. The University of Queensland Animal Experimentation Ethics Committees approved all sampling for echidnas, in accordance with the National Health and Medical Research Council of Australia guidelines 2013.

### Decision letter and Author response

Decision letter https://doi.org/10.7554/eLife.57860.sa1
Author response https://doi.org/10.7554/eLife.57860.sa2

## Additional files

### Supplementary files
• Transparent reporting form

### Data availability
All data generated or analysed during this study are included in the manuscript.

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
