## [Decision Letter]

**Acceptance summary:**

The transformation of mandibles from mammal-like reptiles to crown mammals features the repurposing of morphological structures, as evidenced by integrated molecular, developmental, and fossil data. This article comprises important novel anatomical data from juvenile monotremes and marsupials, presents reasonable interpretations, and will provide a unique insight into the evolution of mammals.

**Decision letter after peer review:**

Thank you for submitting your article "A cranio-incudo joint as the solution to early birth in marsupials and monotremes" for consideration by *eLife*. Your article has been reviewed by three peer reviewers, including Min Zhu as the Reviewing Editor and Reviewer #1, and the evaluation has been overseen by Diethard Tautz as the Senior Editor. The following individuals involved in review of your submission have agreed to reveal their identity: Andrew Gillis (Reviewer #2); Jin Meng (Reviewer #3).

The reviewers have discussed the reviews with one another and the Reviewing Editor has drafted this decision to help you prepare a revised submission.

Summary:

The transformation of mandibles from mammal-like reptiles to crown mammals features the repurposing of morphological structures, as evidenced by integrated molecular, developmental and fossil data. Extensive studies focus on the recurrent origins of the definitive mammalian middle ear, coupled with the shift from the primary jaw joint (quadrate/incus-articular/malleus) to the second jaw joint (squamosal-dentary). Only a few studies deal with the connection between the incus and the basicranium. The study by Anthwal et al. presents their observation on the early stage of the cranio-incudo joint development in monotremes and marsupials; then, by comparing their data with some fossil forms to reach the interpretation/conclusion on the general significance about mammalian jaw evolution.

Overall, this manuscript comprises important novel anatomical data from juvenile monotremes and marsupials, presents reasonable interpretations, and will provide a unique insight into the evolution of mammals. We have the following suggestions to improve the manuscript.

Essential revisions:

1) The authors provide a very thorough description of the comparative anatomy of the Q-A and dentary-squamosal joint in reptiles, mammal-like reptiles and mammals. While this might be very familiar to those with an anatomical background, we fear it might be a bit difficult to follow for many in *eLife*'s broad readership. We recommend the authors add an anatomical schematic to the Introduction, to illustrate the key skeletal articulations and homologies.

2) In the Discussion, the authors might also consider adding a schematic summary of their data, illustrating how the cranio-incudo joint can be observed at the different stages of development in the three mammalian lineages, perhaps including in this schematic an indication of the relevant functional correlates (feeding, etc.)? This would really help to tie the story together.

3) The consistency of term usages throughout the main-text. The craniomandibular (jaw) articulation is conventionally used for the joints between the quadrate and articular (Q-A) and between the squamosal and dentary (temporo-mandibular articulation). Here the authors present the third joint between the quadrate/incus and petrosal, which is well termed in the title (cranio-incudo joint). In the Abstract, the authors suggest the cranio-incudo joint is a subset of the cranio-mandibular articulation (also in the subsection “A double jaw articulation during monotreme development”). As shown in the text, the cranio-incudo joint in juvenile monotremes is just the fusion of two elements. In this case, the joint or the connection might be a proper term. The clarification or consistent usage of these terms will be helpful to the readers.

4) Results section: The authors provide very nice and detailed anatomical descriptions of the embryonic and post-natal conditions in mouse, opossum and echidna/platypus. It would be most helpful for the readers if the authors could open each of the anatomical subsection with a statement of what is currently known (i.e. what is known/speculated to be the case from existing data), what they are aiming to test/verify, and end each subsection with a clear statement of the new findings. Even if these subsections are mainly descriptive, they could still be framed more explicitly as using appropriate taxon sampling and a comparative approach to test a hypothesis of ancestral conditions, function, etc. This would also help to build the story incrementally, in a way that would make it easier for the reader to follow.

---

## [Author Response]

Essential revisions:1) The authors provide a very thorough description of the comparative anatomy of the Q-A and dentary-squamosal joint in reptiles, mammal-like reptiles and mammals. While this might be very familiar to those with an anatomical background, we fear it might be a bit difficult to follow for many in eLife's broad readership. We recommend the authors add an anatomical schematic to the Introduction, to illustrate the key skeletal articulations and homologies.2) In the Discussion, the authors might also consider adding a schematic summary of their data, illustrating how the cranio-incudo joint can be observed at the different stages of development in the three mammalian lineages, perhaps including in this schematic an indication of the relevant functional correlates (feeding, etc.)? This would really help to tie the story together.3) The consistency of term usages throughout the main-text. The craniomandibular (jaw) articulation is conventionally used for the joints between the quadrate and articular (Q-A) and between the squamosal and dentary (temporo-mandibular articulation). Here the authors present the third joint between the quadrate/incus and petrosal, which is well termed in the title (cranio-incudo joint). In the Abstract, the authors suggest the cranio-incudo joint is a subset of the cranio-mandibular articulation (also in the subsection “A double jaw articulation during monotreme development”). As shown in the text, the cranio-incudo joint in juvenile monotremes is just the fusion of two elements. In this case, the joint or the connection might be a proper term. The clarification or consistent usage of these terms will be helpful to the readers.4) Results section: The authors provide very nice and detailed anatomical descriptions of the embryonic and post-natal conditions in mouse, opossum and echidna/platypus. It would be most helpful for the readers if the authors could open each of the anatomical subsection with a statement of what is currently known (i.e. what is known/speculated to be the case from existing data), what they are aiming to test/verify, and end each subsection with a clear statement of the new findings. Even if these subsections are mainly descriptive, they could still be framed more explicitly as using appropriate taxon sampling and a comparative approach to test a hypothesis of ancestral conditions, function, etc. This would also help to build the story incrementally, in a way that would make it easier for the reader to follow.

We thank the reviewers for their helpful comments. We have tried to make all the suggested changes in the text and figures.

In particular we have made the paper clearer for a non-expert reader, adding in new introductory paragraphs, and clarifying knowledge from previous papers to provide a better setting for our study.

As part of this we have added the two suggested schematic figures, to introduce the system (Figure 1) and to highlight our conclusions (Figure 7).

We have clarified the language, to make sure it is clear what we are referring to when discussing the different joints and skeletal elements.

Typos in figures etc have been corrected and the legends made more standardised.

We agree that the title did not really sum up the findings of the paper and so have suggested a new title. We hope this highlights the findings of the paper, which concentrates on the potential role of the incus as a jaw support across mammals.

We have continued to use eutherian rather than placental, as placental suggests that marsupials do not have a placenta during development. This has been explained in the text as we appreciate that placental is more widely used.